# Spectral study of COVID-19 pandemic in Japan: The dependence of spectral gradient on the population size of the community

**Ayako Sumi**[1]*, **Masayuki Koyama**[2,3], **Manato Katagiri**[1,4], **Norio Ohtomo**[5]

**1** Division of Physics, Department of Liberal Arts and Sciences, Center for Medical Education, Sapporo Medical University, Sapporo, Hokkaido, Japan, **2** Department of Public Health, Sapporo Medical University School of Medicine, Sapporo, Hokkaido, Japan, **3** Department of Cardiovascular, Renal and Metabolic Medicine, Sapporo Medical University School of Medicine, Sapporo, Hokkaido, Japan, **4** Sapporo Medical University School of Medicine, Sapporo, Hokkaido, Japan, **5** Natural Energy Research Center Co., Ltd (NERC), Sapporo, Hokkaido, Japan

* sumi@sapmed.ac.jp

**Data Availability Statement:** The dataset of reported COVID-19 cases analyzed during the current study are contained in Supporting

## Abstract

We have carried out spectral analysis of coronavirus disease 2019 (COVID-19) notifications in all 47 prefectures in Japan. The results confirm that the power spectral densities (PSDs) of the data from each prefecture show exponential characteristics, which are universally observed in the PSDs of time series generated by nonlinear dynamical systems, **such as the susceptible/exposed/infectious/recovered (SEIR) epidemic model**. The exponential gradient increases with the population size. For all prefectures, many spectral lines observed in each PSD can be fully assigned to a fundamental mode and its harmonics and subharmonics, or linear combinations of a few fundamental periods, suggesting that the COVID-19 data are substantially noise-free. For prefectures with large population sizes, PSD patterns obtained from segment time series behave in response to the introduction of public and workplace vaccination programs as predicted by theoretical studies based on the SEIR model. The meaning of the relationship between the exponential gradient and the population size is discussed.

## Introduction

Nonlinear dynamics is a simple but powerful framework for modeling and predicting the spread of diseases. The concepts of equilibria, bifurcations, oscillations and chaos provide a natural toolkit for analyzing the spatiotemporal dynamics of transmissible diseases such as influenza [1], acquired immunodeficiency syndrome (AIDS) [2], measles [3], and most recently, coronavirus disease 2019 (COVID-19) [4–6]. One particularly useful technique for detecting and extracting prevalent temporal patterns in nonlinear time series is power spectral density (PSD) calculation. Among other things, the PSDs of time series data allow one to determine the extent to which noise and chaos play roles in their evolutionary dynamics. For instance, in a previous study [7], we confirmed that the PSDs of reported COVID-19 cases in

information files (S1 Dataset). The data are also available from ref. [12].

**Funding:** Grant Number JP22K10529 JSPS KAKEHI. The funder (JSPS KAKENHI) had no role in study design, data collection and analysis, decision to publish, or preparation of the manuscript.

**Competing interests:** The authors have declared that no competing interests exist.

Japan exhibit exponential characteristics, akin to those we observed in the incidence data of measles [8] and time series generated from nonlinear dynamical systems such as the susceptible/exposed/infectious/recovered (SEIR) epidemic model [9]. This suggests that the reported data of new positive cases of COVID-19 in Japan are predominantly influenced by deterministic nonlinear dynamics as opposed to stochasticity. Because deterministic systems are, at least theoretically, predictable, this insight may help scientists and policymakers better understand and control the spread of COVID-19 and similar diseases.

An interesting phenomenon revealed in our prior work is that the gradients of exponential PSDs for time series data of measles notifications increase with population size for communities in the UK, USA, and Denmark [10]. This phenomenon arises because of the growth of spectral power in the low-frequency region and can be attributed to a period-doubling bifurcation process. However, with few exceptions [11], it remains unknown to what extent this trend is present in the data of non-Western countries, such as Japan, where differences in culture and policy may influence the transmission of diseases, and of other diseases, for which methods of transmission may vary wildly. By extending this analysis to COVID-19 notification data collected from the 47 prefectures of Japan, we can gain a better understanding of the mechanisms underlying the transmission of the COVID-19 pandemic in each prefecture as well as the general mechanisms of transmission that are common to all diseases.

Thus, in this study, we investigate whether the prefecture-level PSDs of COVID-19 notification data from the 47 prefectures of Japan exhibit exponential (i.e., nonlinear dynamical) characteristics, whether there exists a correlation between the spectral gradient and the population size of each prefecture, and whether there is evidence of a period-doubling bifurcation that drives the dynamics. Through a combination of the maximum entropy method (MEM) in the frequency domain and the least squares method (LEM) in the time domain, we analyze the fundamental modes of the spectral data and find that the answer to each question is affirmative, implying that much of the data could be explained by a deterministic dynamical model such as a SEIR model. Our examination of these data significantly enhances our understanding of the dynamics of COVID-19 transmission at a regional level in Japan and could aid in refining predictive models for effective pandemic management strategies across the globe.

## Materials and methods

### Data

In Japan, the Omicron strain of COVID-19 was first reported on November 30, 2021. In the present study, we analyze the daily data of reported COVID-19 cases from all 47 prefectures in Japan from 1st February 2022 to 8 May 2023 (714 data points). The first day of the data (February 1, 2022) corresponds to when the number of positive patients with the Omicron strain began to be reported continuously in all 47 prefectures, and the last day of the data (May 8, 2022) corresponds to when the daily data reporting ended. During this period, a total 30,727,445 cases of COVID-19 were reported in Japan. The data were obtained from the Japanese Ministry of Health, Labour and Welfare COVID-19 Data [12] and are listed in S1 Dataset. The 47 prefectures of Japan are shown in Fig 1.

Regarding vaccines administered during the period of data analyzed in this study, the initial vaccination efforts in Japan began in February 2021, focusing first on healthcare workers and the elderly aged 65 and above. By 2024, the first-dose vaccination rate had reached 80.4% [13], while the second and third doses were administered to bolster immunity against the spread of the Alpha and Delta variants. These booster doses were crucial in maintaining protection as new variants emerged [13].

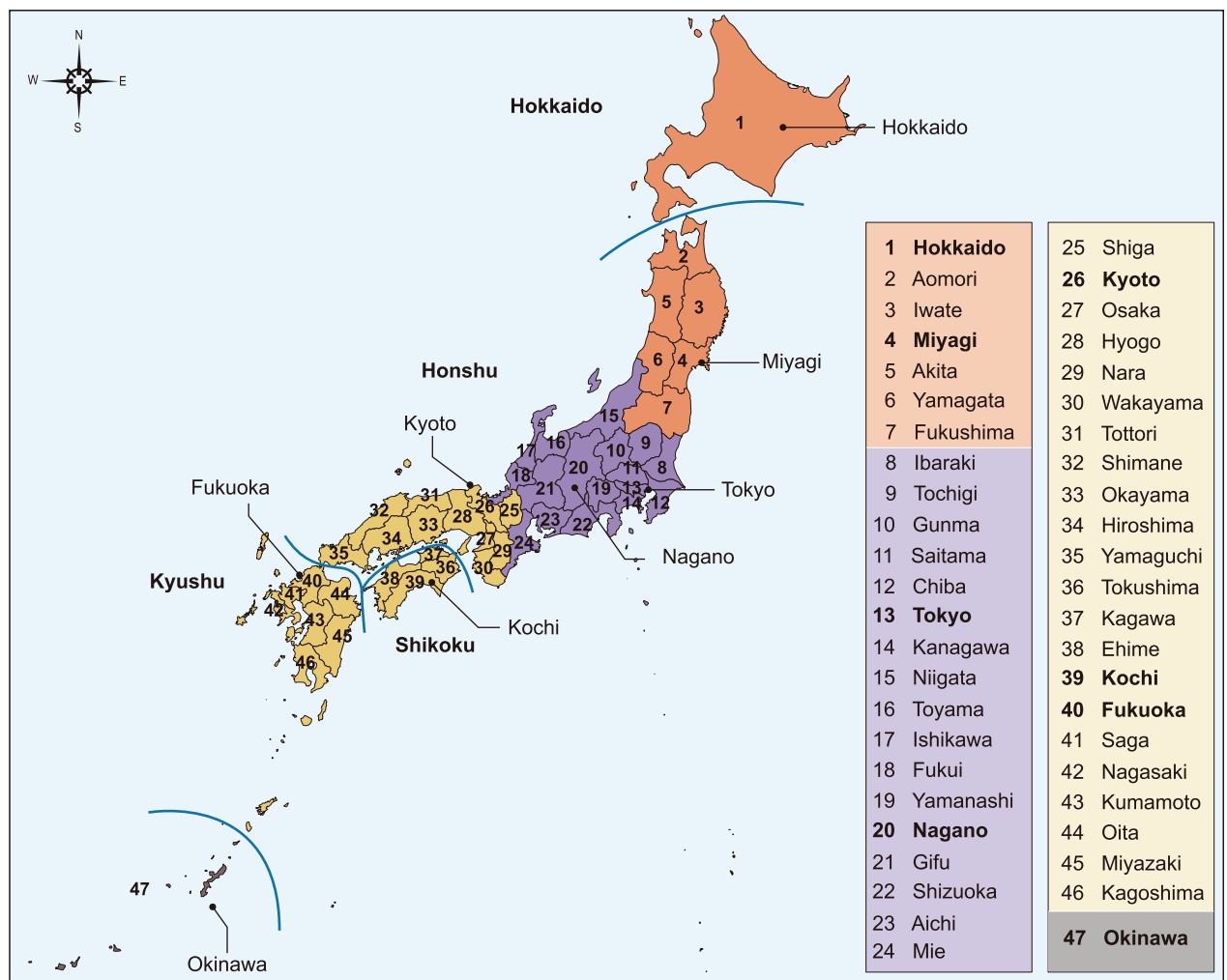

**Fig 1. Distribution of 47 prefectures in Japan.** Dashed lines indicate boundaries of the four main islands constituting Japan: Hokkaido, Honshu, Shikoku, and Kyushu. The eight labeled prefectures are representative sites: (a) Hokkaido and Miyagi in northern Japan (orange); (b) Tokyo and Nagano in eastern Japan (purple); (c) Kyoto, Kochi, and Fukuoka in western Japan (yellow); and (d) Okinawa in southern Japan (gray).

The fourth dose of the COVID-19 vaccine, initiated on May 25, 2022, primarily targeted individuals aged 60 and older along with those with underlying health conditions, achieving a vaccination rate of approximately 54% by early 2023 [13]. Meanwhile, the bivalent Omicron-adapted vaccine, introduced in the autumn of 2022, was rapidly expanded to broader age groups, resulting in a total of around 28.4 million doses that had been administered by 2023 [14].

## Selection method of eight prefectures as representative sites

Based on the geographical division of the Japan Meteorological Agency [15], we selected eight prefectures from all 47 prefectures in Japan as illustrative examples. The Japan Meteorological Agency divides the entire country into the following four areas for weather services and information: northern Japan, eastern Japan, western Japan and southern Japan. In this study, we selected one to three prefectures from each area as illustrative examples (Fig 1) as follows: (a) Hokkaido and Miyagi from northern Japan; (b) Tokyo and Nagano from eastern Japan; (c)

Kyoto, Kochi and Fukuoka from western Japan; and (d) Okinawa from southern Japan. This gave us a total of eight prefectures. Japan consists of four main islands (Honshu, Shikoku, Kyushu, Hokkaido). Seven of the above selected prefectures (excluding Okinawa) were chosen to cover the four main islands that constitute the country: Honshu (Miyagi, Tokyo, Nagano, Kyoto), Shikoku (Kochi), Kyushu (Fukuoka) and Hokkaido (Hokkaido).

## Time series analysis

Our time series analysis consists of maximum entropy method (MEM) spectral analysis in the frequency domain and the least squares method (LSM) in the time domain [16]. The MEM is considered to have a high degree of resolution of spectral estimates compared with other methods of analyzing infectious disease surveillance data such as the fast Fourier transform algorithm and autoregressive methods, which require time series with long data lengths [16–19]; therefore, an MEM spectral analysis allows us to precisely determine short data sequences, such as the infectious disease surveillance data used in this study. The method used for analysis in this study can be applied to any time series with at least seven data points [20, 21]. For example, it has been applied to time series data of sunspot numbers [22], which were used as a test case for time series analysis, and its usefulness has been confirmed.

**MEM spectral analysis.** We assume that the time series $x(t)$ (where $t$ is time) is composed of systematic and fluctuating parts [23]:

$$x(t) = \text{systematic part} + \text{fluctuating part}. \tag{1}$$

To investigate temporal patterns of $x(t)$ in the monthly time series data, we performed MEM spectral analysis [7–10, 16–20]. MEM spectral analysis produces a PSD, from which we obtain the power representing the amount of amplitude of $x(t)$ at each frequency (note the reciprocal relationship between frequency and period). The so-called MEM-PSD, $P(f)$ (where $f$ represents frequency), for the time series with uniform sampling interval $\Delta t$, can be expressed as

$$P(f) = \frac{P_m \Delta t}{\left| 1 + \sum_{k=-m}^{m} \gamma_{m,k} \exp[-i2\pi f k \Delta t] \right|^2}, \tag{2}$$

where the value of $P_m$ is the output power of a prediction–error filter of order $m$ and $\gamma_{m,k}$ is the corresponding filter order. The value of the MEM-estimated period of the $n$th peak component $T_n$ ($= 1/f_n$; where $f_n$ is the frequency of the $n$th peak component) can be determined by the positions of the peaks in MEM-PSD. The correlation of the gradient of the PSD for COVID-19 time series data against population size for all 47 prefectures was evaluated via Pearson's correlation ($\rho$) analysis performed using Statistical Package for the Social Sciences (SPSS) version 17.0J software (SPSS, Japan). A $p$ value of $\leq 0.05$ was considered statistically significant.

**LSM.** The validity of the MEM spectral analysis results was confirmed by calculating the least squares fitting (LSF) curve pertaining to the original time series data $x(t)$ with MEM-estimated periods. The formula used to generate the LSF curve $X(t)$ was as follows:

$$X(t) = A_0 + \sum_{n=1}^{N} A_n \cos\{2\pi f_n(t + \theta_n)\}. \tag{3}$$

The above formula is calculated using the LSM for $x(t)$ with unknown parameters $f_n$, $A_0$ and $A_n$ ($n = 1, 2, 3, \ldots, N$), where $f_n$ ($= 1/T_n$; $T_n$ is the period) is the frequency of the $n$-th

component, $A_0$ is a constant that indicates the average value of the time series data, $A_n$ is the amplitude of the $n$-th component, $\theta_n$ is the phase of the $n$-th component, and $N$ is the total number of components.

**Preparing the data for analysis.** Fig 2a–2h indicates the daily reported number of new positive cases of COVID-19 from 1 February 2022 to 8 May 2023, which we denote $x(t)$, for the following eight prefectures: (a) Hokkaido, (b) Miyagi, (c) Tokyo, (d) Nagano, (e) Kyoto, (f) Kochi, (g) Fukuoka and (h) Okinawa. Fig 2a'–2h' shows the frequency histogram for $x(t)$ (Fig 2a–2h, respectively). Each histogram deviates from the normal distribution required for conventional spectral analysis. We introduced a logarithmic transformation of $x(t)$ (Fig 2a–2h). Logarithm-transformed data for this period are shown in Fig 3a–3h. The frequency histograms of the logarithm-transformed data in Fig 3a–3h are shown in Fig 3a'–3h', respectively, and approximately obey the normal distribution required for conventional spectral analysis, although a slight difference was observed.

**Segment time series analysis.** We further investigated periodic structures of the logarithm-transformed data (Fig 3a–3h) by using segment time series analysis. The logarithm-transformed data (Fig 3a–3h) were divided into several segments and the MEM-PSD for each segment was computed. In this study, each segment represents a 380-day time interval, with the starting points of two consecutive segments separated by 1 day. The obtained MEM-PSDs were arranged in the order of the time sequence to construct a three-dimensional (3D) spectral array.

## Results

### Temporal variations of reported number of new positive COVID-19 cases

The original data for the eight prefectures (Fig 2a–2h) exhibited two large peaks: one in the summer during August 2022 and one in the winter during November 2022 (for Hokkaido and

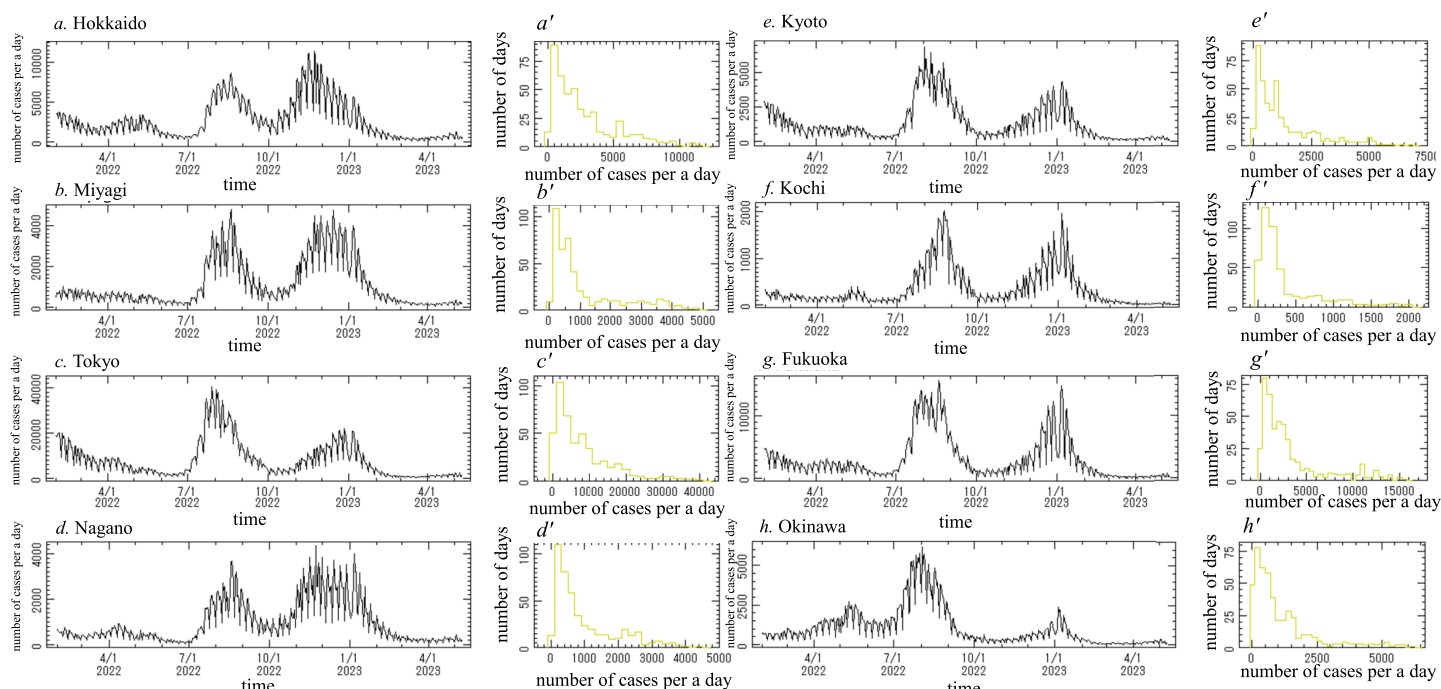

**Fig 2. Original time series data of daily reported number of new positive COVID-19 cases for eight prefectures in Japan from 1 February 2022 to 8 May 2023 (a-h) and their histograms (a'-h').** (a) and (a') for Hokkaido, (b) and (b') for Miyagi, (c) and (c') for Tokyo, (d) and (d') for Nagano, (e) and (e') for Kyoto, (f) and (f') Kochi, (g) and (g') for Fukuoka, and (h) and (h') for Okinawa.

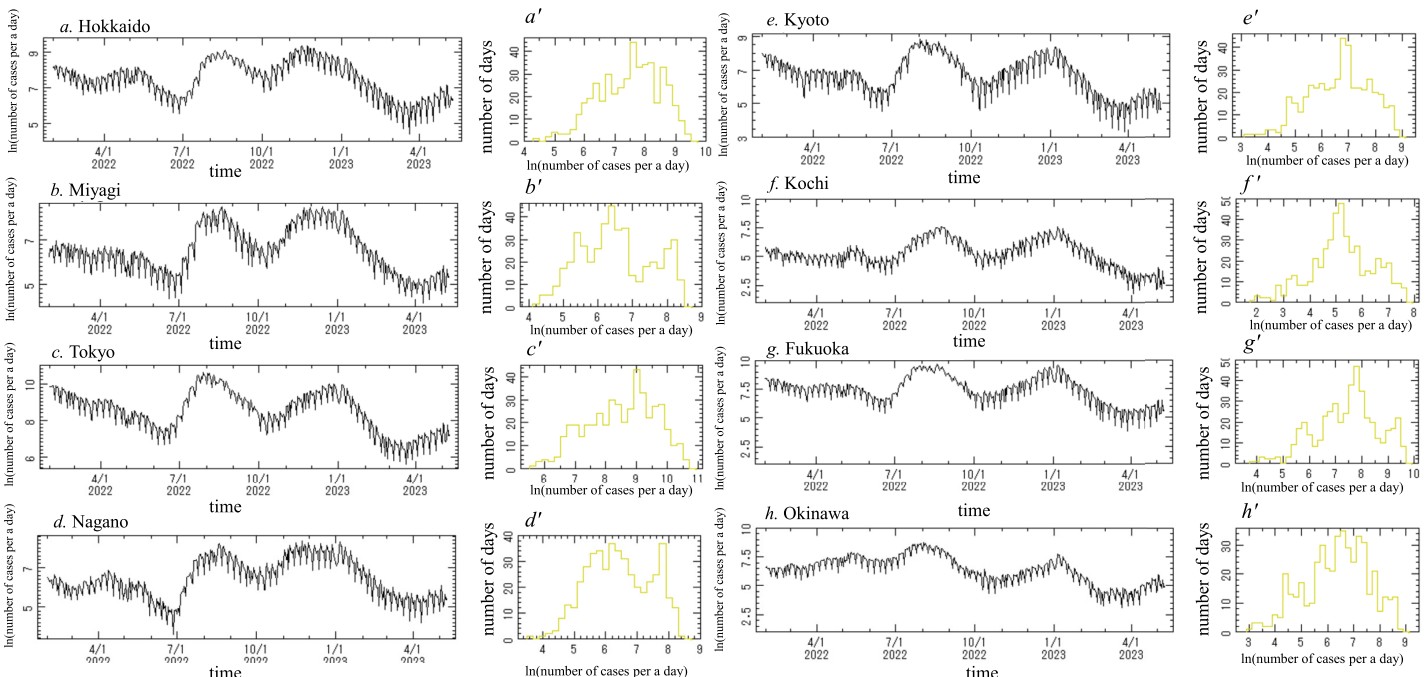

**Fig 3. Logarithm-transformed data of the original data (Fig 2a–2h) and histograms of the logarithm-transformed data.** (a) and (a') for Hokkaido, (b) and (b') for Miyagi, (c) and (c') for Tokyo, (d) and (d') for Nagano, (e) and (e') for Kyoto, (f) and (f') Kochi, (g) and (g') for Fukuoka, and (h) and (h') for Okinawa.

Nagano), December 2022 (for Miyagi and Tokyo), or January 2023 (for Kyoto, Kochi, Fukuoka and Okinawa).

## Gradient of PSD

MEM-PSDs for the logarithm-transformed data of eight prefectures (Fig 3a–3h) were calculated. The semi-log plots of the PSDs are shown in Fig 4a–4h ($f < 50.0$) (unit of frequency: 1/year). Broad continuous spectra appear and many well-defined spectral lines superposed on the spectrum are clearly observed as dominant peaks. The overall trends of the PSDs for all prefectures suggest the exponential form

$$P(f) = \exp(-\lambda f) \tag{4}$$

until the PSDs level off at the lowest limit determined by the accuracy of the present data, that is, the number of significant digits in the data. To obtain the magnitude of $\lambda$, the mean power of the PSD was calculated by integrating the PSD over a small frequency interval $\Delta f$, that is, the mean power of the PSD is the power in the interval of frequencies $[f, f + \Delta f]$. The line of the PSD gradient is calculated as a regression line against the mean powers in the low frequency range ($f < 12.0$), as illustrated in Fig 4. The precise value of $\lambda$ was determined using this procedure. We find that $\lambda = 0.29, 0.28, 0.32, 0.24, 0.29, 0.25, 0.35,$ and $0.22$ for Hokkaido, Miyagi, Tokyo, Nagano, Kyoto, Kochi, Fukuoka, and Okinawa, respectively.

The exponential spectrum and broad continuous spectrum observed for the eight prefectures are also observed for the other 39 prefectures, as shown in S1 Fig. The values of $\lambda$ for all 47 prefectures are plotted in Fig 5 against the logarithm-transformed population size of each prefecture. The value of $\lambda$ apparently increases as the population size increases, and $\lambda$ is

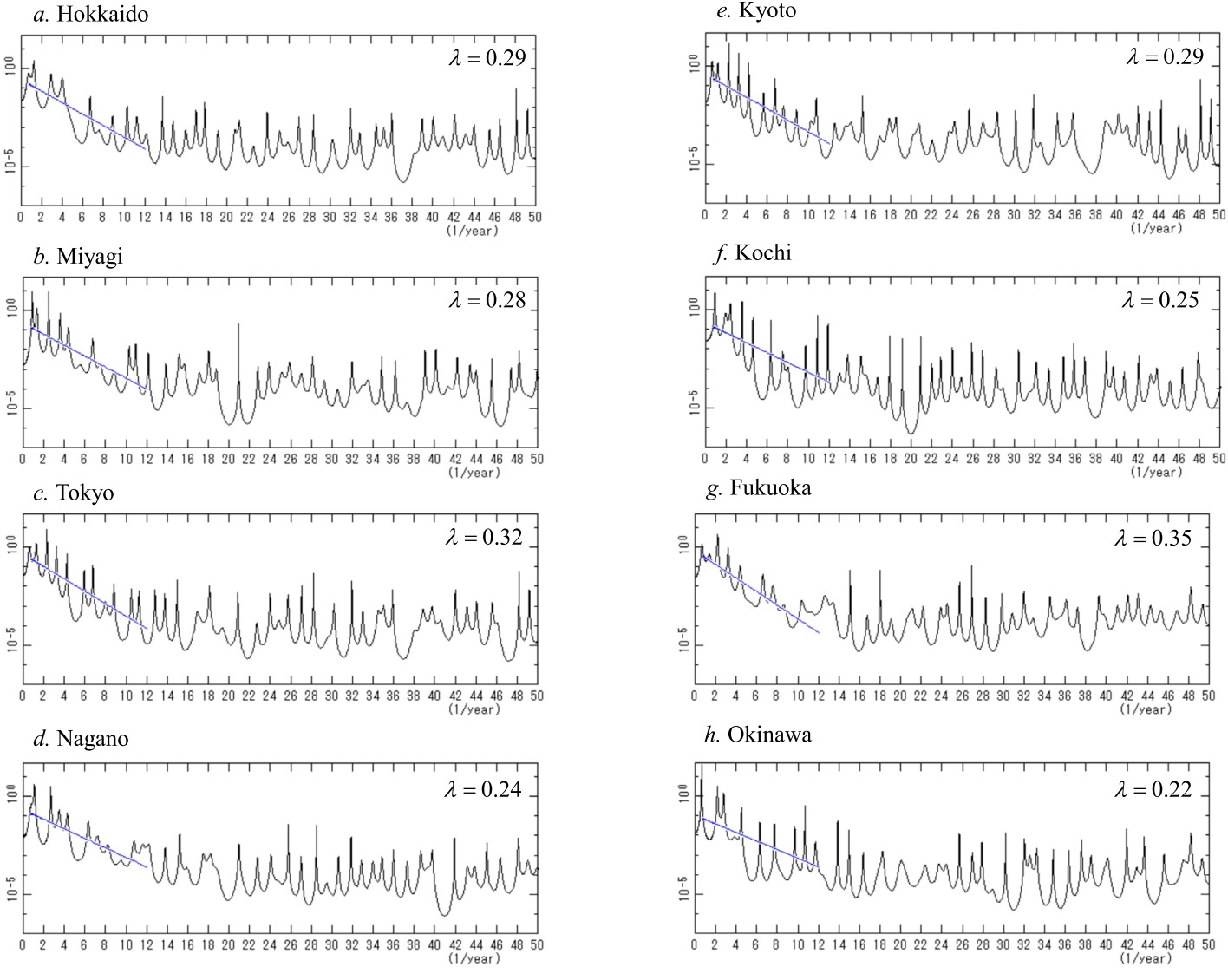

**Fig 4. MEM-PSDs for the logarithm-transformed data ($f < 50.0$).** (a) Hokkaido, (b) Miyagi, (c) Tokyo, (d) Nagano, (e) Kyoto, (f) Kochi, (g) Fukuoka, and (h) Okinawa. In each figure, the blue line in the low-frequency range ($f < 12.0$) indicates the regression line with the gradient of the MEM-PSD, $\lambda$, of Eq (4). See the text for details.

significantly correlated with population size ($\rho = 0.47$, $p < 0.01$). The linear regression line is drawn in the figure, and the value of slope is 0.0454 (the correlation coefficient, $R^2$: 0.185).

Fig 5 also shows the relationship between the two durations. $K$-means clustering ($n = 2$) was applied to separate the data into two major groups; Cluster 1 and Cluster 2. Among all 47 prefectures, 10 prefectures are assigned to Cluster 1, which consists of prefectures with large population sizes (3,553,000–14,100,000 people), and the other 37 prefectures are assigned to Cluster 2, which consists of prefectures with small population sizes (537,000–2,826,000 people). For the eight representative sites (Fig 5), three prefectures (Hokkaido, Tokyo, and Fukuoka) are in Cluster 1, and the other five prefectures (Miyagi, Nagano, Kyoto, Kochi, and Okinawa) are in Cluster 2. The $\lambda$ values for each prefecture are listed in Table 1, where all 47 prefectures are arranged in order of population size.

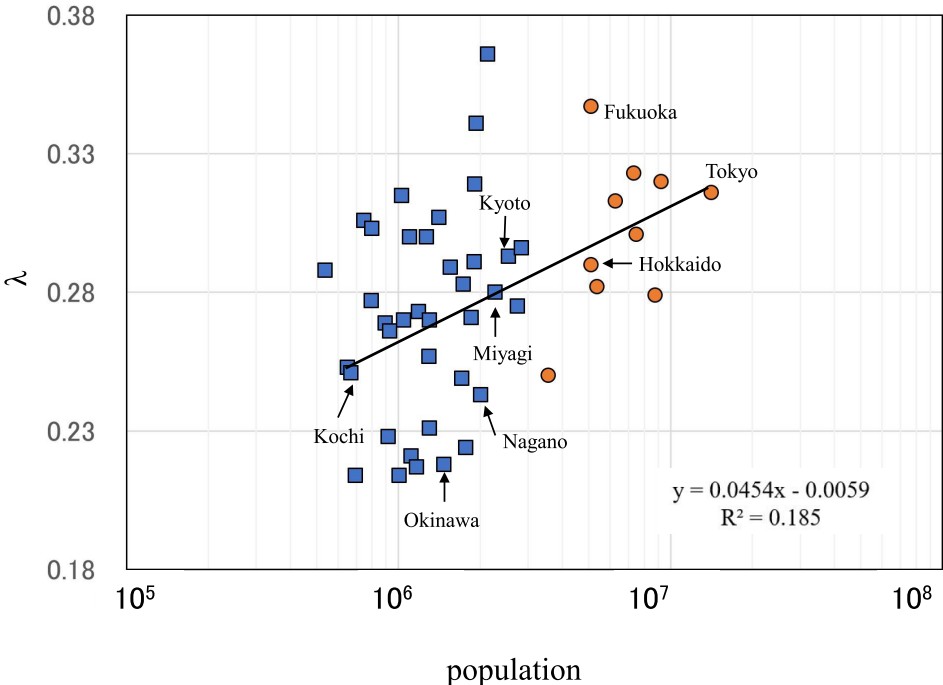

**Fig 5. Gradients of MEM-PSDs ($\lambda$) against population size of 47 prefectures in Japan.** Blue squares indicate prefectures in Cluster 1 with large population sizes (3,553,000–14,100,000 people) and orange circles indicate prefectures in Cluster 2 with small population sizes (537,000–2,826,000 people). The linear regression line for $\lambda$ is drawn as a solid line.

## Dominant spectral lines

A close-up of the low-frequency ranges of the PSDs in Fig 4a–4h are shown in Fig 6a–6h ($f < 10.0$). Therein, many well-defined spectral lines are observed in each PSD.

For all eight prefectures, the most prominent spectral lines are observed at $f_1$ in the frequency range from 2.17 to 2.89, that is, from 126.3 to 168.2 days: $f$ = 2.89, 2.47, 2.29, 2.67, 2.25, 2.41, 2.20, and 2.17 for Hokkaido, Miyagi, Tokyo, Nagano, Kyoto, Kochi, Fukuoka, and Okinawa, respectively. In the case of Tokyo (Fig 6c), for example, the dominant spectral peak corresponding to the fundamental mode $f_1$ is observed at a frequency of 2.29, corresponding to approximately 159.3 days.

In the PSDs of all eight prefectures, spectral lines of harmonics of $f_1$, that is, $f_2$ ($= f_1 \times 2$) and $f_3$ ($= f_1 \times 3$) are clearly observed, although in the case of Hokkaido (Fig 6a), the peaks of $f_2$ and $f_3$ cannot be recognized at their normal positions.

The dominant spectral lines are classified into one of two patterns based on whether spectral lines of subharmonics $f_{1/2}$ ($= 1/2 \times f_1$) and their harmonics $f_{3/2}$ ($= 3 \times f_{1/2}$) are observed in the PSD. We refer to these patterns as subharmonic patterns and non-subharmonic patterns, respectively.

**Subharmonic patterns.** In the cases of Miyagi, Tokyo, Kyoto, and Fukuoka (Fig 6b, 6c, 6e and 6g, respectively), spectral lines of subharmonics $f_{1/2}$ and their harmonics $f_{3/2}$ are observed. The result of the assignment of spectral peak frequencies for Tokyo, for example is shown in Table 2a.

**Non-subharmonic patterns.** Regarding the cases of Hokkaido, Nagano, Kochi, and Okinawa (Fig 6a, 6d, 6f and 6h, respectively), spectral lines of $f_{1/2}$ are not observed, and almost all

**Table 1. Population size and results of COVID-19 notification data.**

|  | Prefecture | Population size | λ | Fundamental mode (days) | Pattern |
|---|---|---|---|---|---|
| Cluster 1 | Tokyo | 14,099,993 | 0.32 | 159.3 | subharmonic |
|  | Kanagawa | 9,229,713 | 0.32 | 157.9 | subharmonic |
|  | Osaka | 8,774,574 | 0.28 | 164.4 | subharmonic |
|  | Aichi | 7,480,897 | 0.30 | 163.8 | subharmonic |
|  | Saitama | 7,331,296 | 0.32 | 158.2 | subharmonic |
|  | Chiba | 6,273,530 | 0.31 | 162.4 | subharmonic |
|  | Hyogo | 5,369,834 | 0.28 | 165.2 | subharmonic |
|  | Fukuoka | 5,106,912 | 0.35 | 166.1 | subharmonic |
|  | Hokkaido | 5,091,680 | 0.29 | 300.2, 126.3 | non-subharmonic |
|  | Shizuoka | 3,553,518 | 0.25 | 161.5 | subharmonic |
| Cluster 2 | Ibaraki | 2,826,047 | 0.30 | 298.0, 153.6 | non-subharmonic |
|  | Hiroshima | 2,739,446 | 0.28 | 357.8, 133.7 | non-subharmonic |
|  | Kyoto | 2,536,995 | 0.29 | 162.2 | subharmonic |
|  | Miyagi | 2,263,552 | 0.28 | 148.1 | subharmonic |
|  | Niigata | 2,126,276 | 0.37 | 344.3, 141.5 | non-subharmonic |
|  | Nagano | 2,004,785 | 0.24 | 338.0, 136.7 | non-subharmonic |
|  | Gifu | 1,929,669 | 0.34 | 158.0 | subharmonic |
|  | Gunma | 1,900,840 | 0.32 | 153.4 | subharmonic |
|  | Tochigi | 1,895,031 | 0.29 | 334.9, 147.2 | non-subharmonic |
|  | Okayama | 1,846,525 | 0.27 | 347.6, 183.4 | non-subharmonic |
|  | Fukushima | 1,766,358 | 0.22 | 321.3, 137.3 | non-subharmonic |
|  | Mie | 1,727,503 | 0.28 | 160.1 | subharmonic |
|  | Kumamoto | 1,707,747 | 0.25 | 263.5, 169.1 | non-subharmonic |
|  | Kagoshima | 1,547,710 | 0.29 | 180.6 | non-subharmonic |
|  | Okinawa | 1,468,375 | 0.22 | 168.0 | non-subharmonic |
|  | Shiga | 1,406,103 | 0.31 | 160.7 | subharmonic |
|  | Yamaguchi | 1,296,593 | 0.27 | 427.9, 164.3 | non-subharmonic |
|  | Nara | 1,295,681 | 0.23 | 163.7 | subharmonic |
|  | Ehime | 1,291,198 | 0.26 | 422.5, 161.1 | non-subharmonic |
|  | Nagasaki | 1,266,334 | 0.30 | 452.9, 161.8 | non-subharmonic |
|  | Aomori | 1,184,531 | 0.27 | 299.4, 144.3 | non-subharmonic |
|  | Iwate | 1,163,024 | 0.22 | 252.2, 146.2 | non-subharmonic |
|  | Ishikawa | 1,109,574 | 0.22 | 319.3, 152.5 | non-subharmonic |
|  | Oita | 1,096,235 | 0.30 | 392.1, 150.4 | non-subharmonic |
|  | Miyazaki | 1,041,150 | 0.27 | 452.9, 165.4 | non-subharmonic |
|  | Yamagata | 1,026,228 | 0.32 | 357.8, 134.7 | non-subharmonic |
|  | Toyama | 1,006,367 | 0.21 | 384.2, 154.3 | non-subharmonic |
|  | Kagawa | 925,408 | 0.27 | 372.8, 150.9 | non-subharmonic |
|  | Akita | 913,556 | 0.23 | 324.7, 131.6 | non-subharmonic |
|  | Wakayama | 891,620 | 0.27 | 183.4 | subharmonic |
|  | Yamanashi | 795,544 | 0.30 | 351.0, 149.0 | non-subharmonic |
|  | Saga | 794,385 | 0.28 | 261.5, 167.0 | non-subharmonic |
|  | Fukui | 744,568 | 0.31 | 153.4 | subharmonic |
|  | Tokushima | 694,841 | 0.21 | 397.2, 169.4 | non-subharmonic |
|  | Kochi | 666,293 | 0.25 | 405.6, 151.5 | non-subharmonic |
|  | Shimane | 649,235 | 0.25 | 422.9, 161.6 | non-subharmonic |
|  | Tottori | 537,318 | 0.29 | 404.7, 153.0 | non-subharmonic |

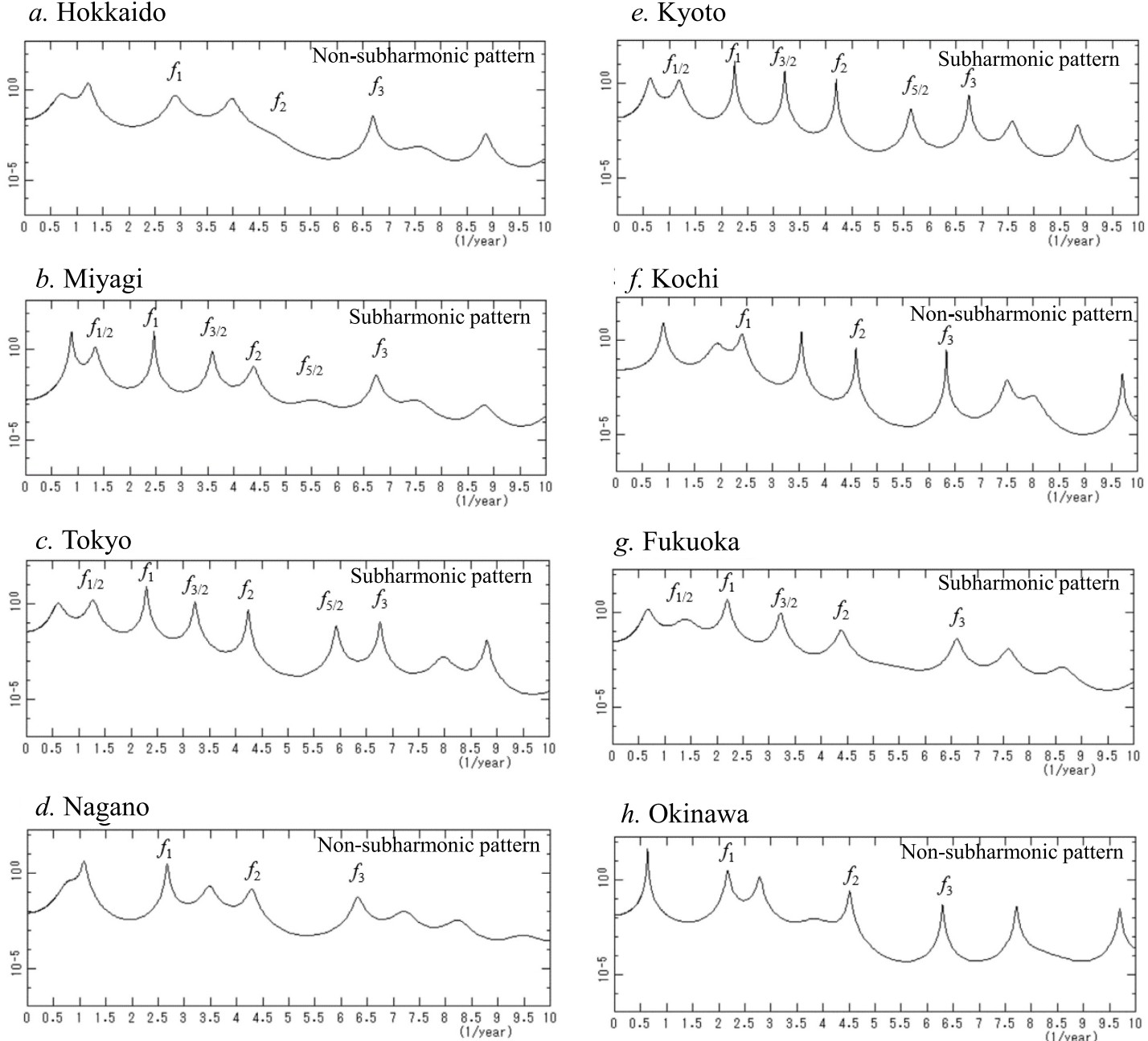

**Fig 6. Close-up of the low-frequency region ($f < 10.0$) in Fig 4.** (a) Hokkaido, (b) Miyagi, (c) Tokyo, (d) Nagano, (e) Kyoto, (f) Kochi, (g) Fukuoka, and (h) Okinawa.

spectral peak frequencies observed in each PSD can be assigned to linear combinations of a few fundamental modes corresponding to the frequencies of dominant spectral lines observed in the PSD. The result of the assignment for Hokkaido, for example, is shown in Table 2b. Therein, frequencies $f_I = 1.22$ and $f_{II} = 2.89 \, (= f_1)$ were assigned, and the combinations of these frequencies could be used to completely interpret almost all the spectral lines, except for the lowest frequency line which was affected by the length of data. An outline of the assignment process for spectral peak frequencies is described in S1 Appendix.

**Table 2. Assignment of the first seven spectral peaks ($f_{obs}$) observed in Fig 6c using the relation $f_{theory} = |m_I f_I + m_{II} f_{II}|$ ($m_I$, $m_I$: Positive integer; $f_I$, $f_{II}$: The fundamental mode).** —: not assigned.

*a.* Tokyo

$f_I = 159.3$ (day)

| $f_{obs}$ | (period[day]) | power | $f_{theory}$ | $m_I$ | $\Delta f = |f_{theory} - f_{obs}|$ | $\Delta f / f_{theory}$ |
|---|---|---|---|---|---|---|
| 0.604 | 604.3 | 0.296 | - - - | - - - | - - - | - - - |
| 1.268 | 287.9 | 0.311 | 1.146 | a) | 0.122 | 0.1065 |
| 2.292 | 159.3 | 0.411 | 2.292 | 1 | 0.000 | 0.0000 |
| 3.223 | 113.2 | 0.094 | 3.438 | b) | 0.215 | 0.0625 |
| 4.24 | 86.1 | 0.025 | 4.584 | 2 | 0.344 | 0.0750 |
| 5.927 | 61.6 | 0.006 | 5.73 | c) | 0.197 | 0.0344 |
| 6.764 | 54.0 | 0.008 | 6.876 | 3 | 0.112 | 0.0163 |

*b.* Hokkaido

$f_I = 300.2$ (day)
$f_{II} = 126.3$ (day)

| $f_{obs}$ | (period [days]) | power | $f_{theory}$ | $m_I$ | $m_{II}$ | $\Delta f = |f_{theory} - f_{obs}|$ | $\Delta f / f_{theory}$ |
|---|---|---|---|---|---|---|---|
| 0.710 | 514.1 | 0.20441 | - - - | - - - | - - - | - - - | - - - |
| 1.216 | 300.2 | 0.42263 | 1.22 | 1 | - - - | 0.000 | 0.0000 |
| 2.890 | 126.3 | 0.14191 | 2.89 | — | 1 | 0.000 | 0.0000 |
| 3.982 | 91.7 | 0.09590 | 4.11 | 1 | 1 | 0.124 | 0.0302 |
| 6.693 | 54.5 | 0.00420 | 5.78 | - - - | 2 | 0.913 | 0.1580 |
| 7.564 | 48.3 | 0.00045 | 7.00 | 1 | 2 | 0.568 | 0.0812 |
| 8.866 | 41.2 | 0.00072 | 8.67 | - - - | 3 | 0.196 | 0.0226 |

a) subharmonics ($1/2 \times f_I$) of the fundamental mode $f_I$,

b) and c) its odd harmonics ($3/2 \times f_I$ and $5/2 \times f_I$, respectively).

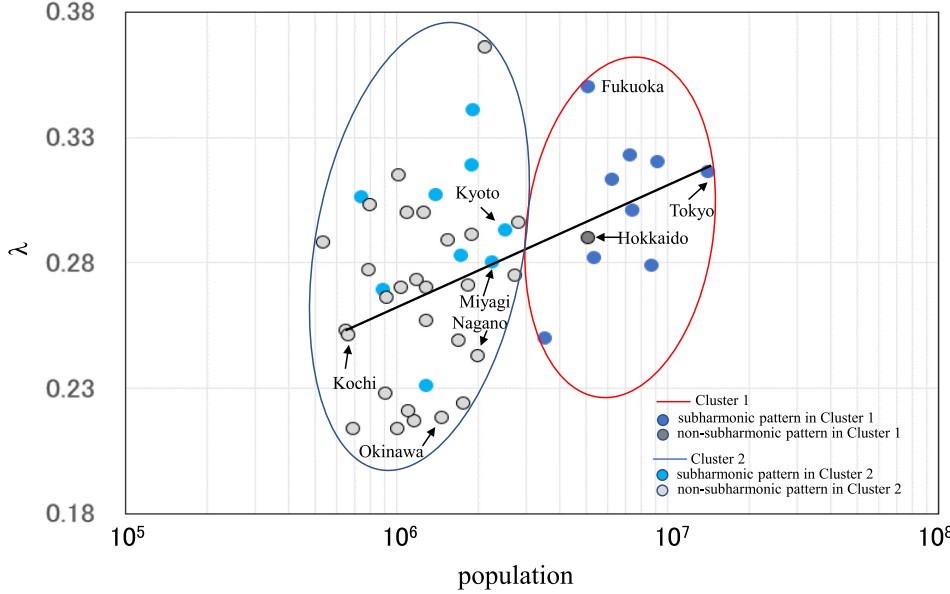

**Fig 7. Relationship between the results of the clustering (Fig 5) and the dominant spectral lines of the PSDs (Fig 6).** Prefectures in Cluster 1 (Fig 5) are surrounded by a red line and prefectures in Cluster 2 (Fig 5) are surrounded by a blue line. In Cluster 1 (resp. Cluster 2), prefectures with subharmonic patterns are indicated by a dark blue (resp. light blue) cycle and prefectures with non-subharmonic patterns are indicated by a dark gray (resp. light gray) cycle.

The PSDs of the other 39 prefectures are also classified as subharmonic patterns or non-subharmonic patterns, and the results are listed in Table 1. The result that all spectral peak frequencies are explained by either one fundamental mode (subharmonics pattern) or multiple fundamental modes (non-subharmonics pattern) holds for all 47 prefectures, indicating that the assigned spectral peaks are not due to noise.

An alternate version of Fig 5 is shown in Fig 7, which shows the relationship between the results of the clustering (Fig 5) and the dominant spectral lines of the PSDs (Fig 6). For Cluster 1 (large population sizes) in Fig 7, nine out of ten of the prefectures (90%) are belong to the subharmonic pattern category. Conversely, for Cluster 2 (small population sizes) in Fig 7, the patterns are predominantly non-subharmonic, with 28 of 37 prefectures (76%) having patterns classified as non-subharmonic and the remaining 9 (24%) as subharmonic.

Fig 8 shows the spatial distribution of prefectures showing subharmonic and non-subharmonic patterns. Prefectures showing subharmonic patterns in Cluster 1 are located in the

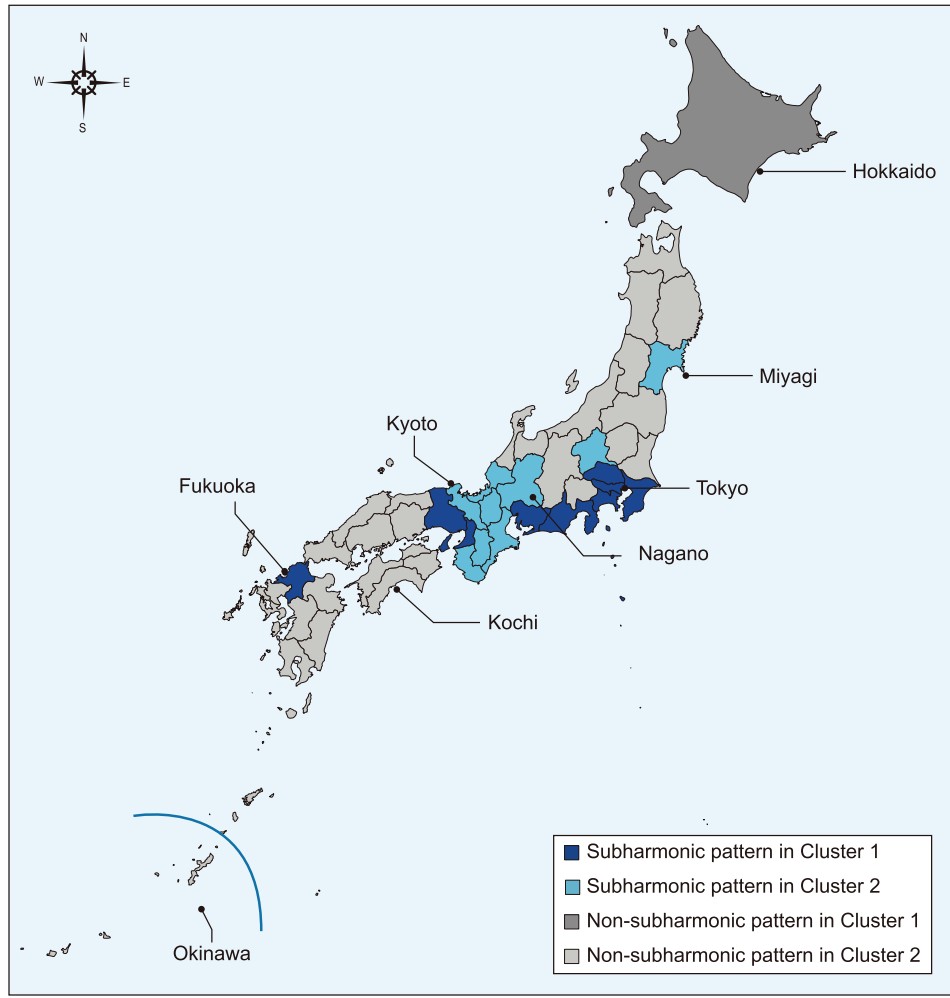

**Fig 8. Spatial distribution of prefectures showing subharmonic and non-subharmonic patterns in Fig 7.**
Prefectures colored dark blue (resp. light blue) possess subharmonic patterns and belong to Cluster 1 (resp. Cluster 2). Prefectures colored dark gray (resp. light gray) possess non-subharmonic patterns and belong to Cluster 1 (resp. Cluster 2).

center of Japan, except for Fukuoka in western Japan. Prefectures showing subharmonic patterns in Cluster 2 are also located in the center of Japan, except for Miyagi in northern Japan.

## Least squares method

The optimal curves obtained by least squares fitting (LSF) of the data were calculated using LSM with the fundamental modes (Table 1). The LSF curves obtained are shown in Fig 9a–9h and are compared with the original time series. As seen in each figure, the LSF curve is essentially in agreement with the original time series, although fine variations superposed on the long-term oscillation cannot be reproduced well. Thus, the fundamental modes assigned (Table 1) were confirmed to be appropriate.

## Segment time series analysis

Three-dimensional spectral arrays for the eight prefectures are shown in Fig 10a–10h, where power is plotted versus frequency (abscissa) and time (right-hand ordinate). For each

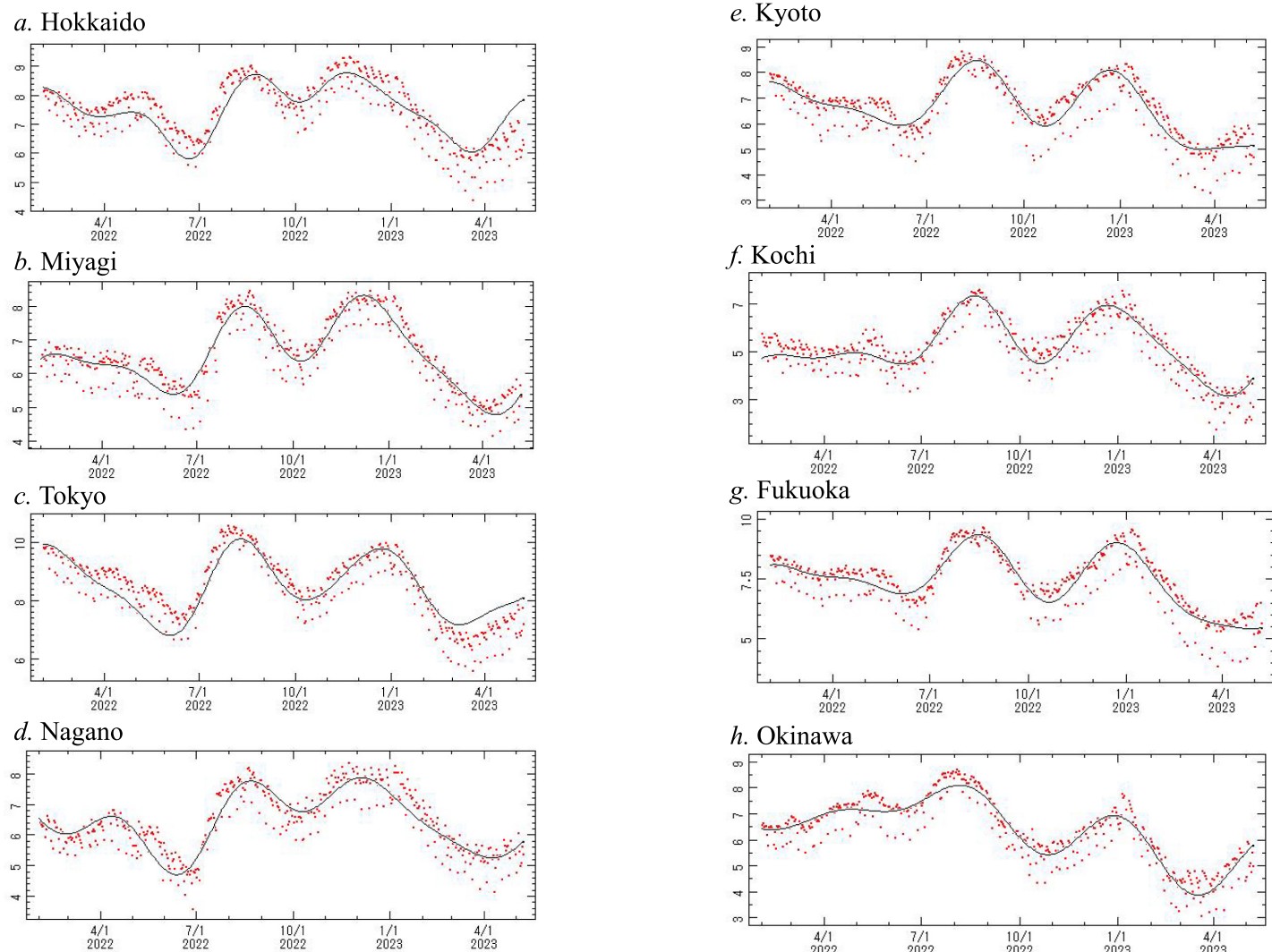

**Fig 9. Comparison of the least squares fitting curve (solid line) with the logarithm-transformed data in Fig 3 (red dot).** (a) Hokkaido, (b) Miyagi, (c) Tokyo, (d) Nagano, (e) Kyoto, (f) Kochi, (g) Fukuoka, and (h) Okinawa.

prefecture, the spectral arrays of $f_{<1}, f_1,$ and $f_2$ are clearly displayed over the entire time range, where $f_{<1}$ denotes the frequency lower than $f_1$.

The temporal variations of spectral lines of $f \leq 1.6$ observed in the 3D spectral arrays (Fig 10a–10h) are plotted in Fig 11 with cumulative numbers of the fourth vaccination, introduced in late-May 2022, and the Omicron compatible bivalent vaccine, introduced in mid-September 2022 [24]. In the cases of Tokyo, Kyoto, and Fukuoka, spectral peaks assigned as $f_{1/2}$ were observed and gradually moved to lower-frequency regions over time, with the rate of movement changing from September 2022.

## Discussion

An important result of this study is that exponential characteristics were observed in the COVID-19 data for 47 prefectures. Exponential spectra are observed in the Rössler and SEIR models, which are well-known nonlinear dynamical models. Therefore, the COVID-19 data of the 47 prefectures have characteristics observed in nonlinear dynamical systems.

The second important result of this study is that, as Fig 11 shows, the spectral peak frequency identified as $f_{1/2}$ becomes larger as the vaccination coverage of the fourth vaccination, introduced in late-May 2022, and Omicron-compatible bivalent vaccine, introduced in mid-September 2022, increased. This fact could be explained by the SEIR model in the same way that the 2-year cycle generated by the period-doubling bifurcations of the SEIR model was reported to explain the 2-year measles cycle. It was theoretically derived and confirmed by surveillance data of measles in Japan and Wuhan in China that the 2-year cycle lengthens with increasing immunization coverage. Our result that the spectral peak frequency identified at $f_{1/2}$ of the COVID-19 data gradually moves to a lower-frequency range, that is, the period of the

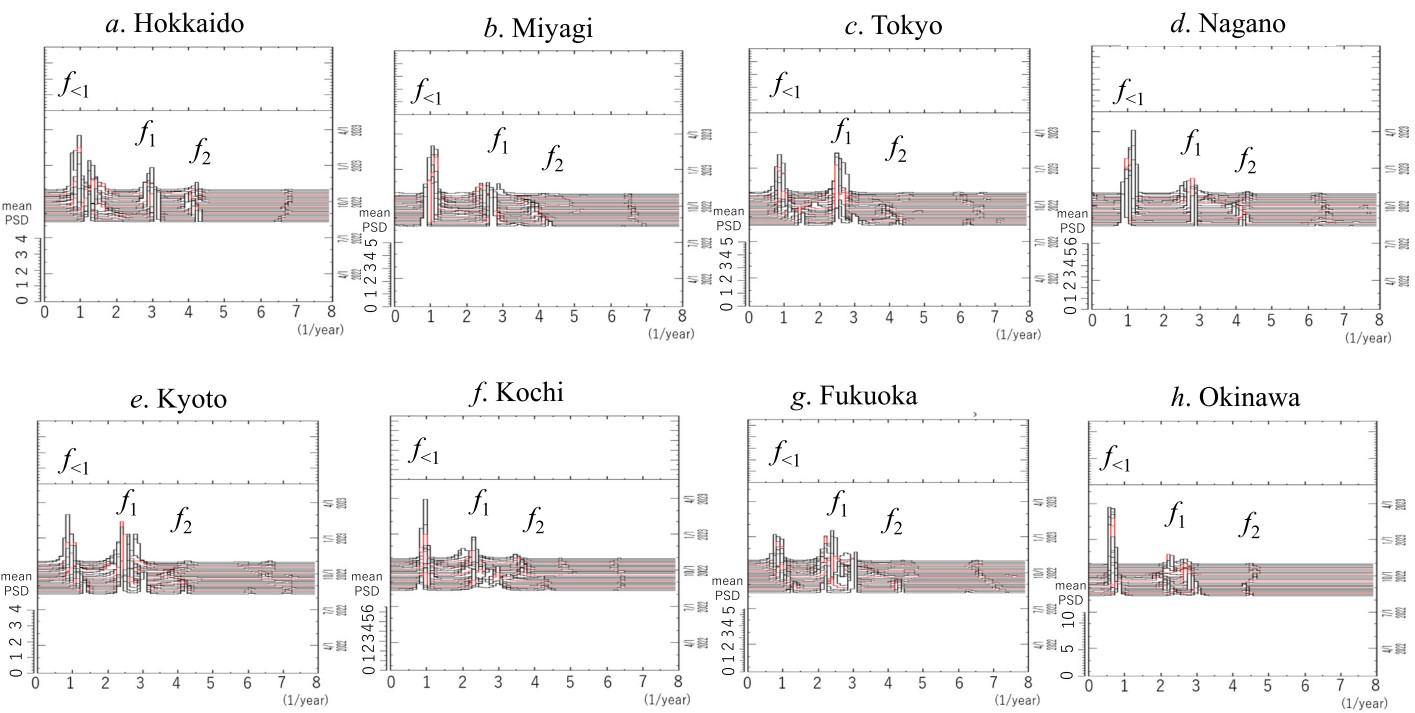

**Fig 10. Three-dimensional spectral arrays for the residual data in the frequency range of $f \leq 8.0$.**

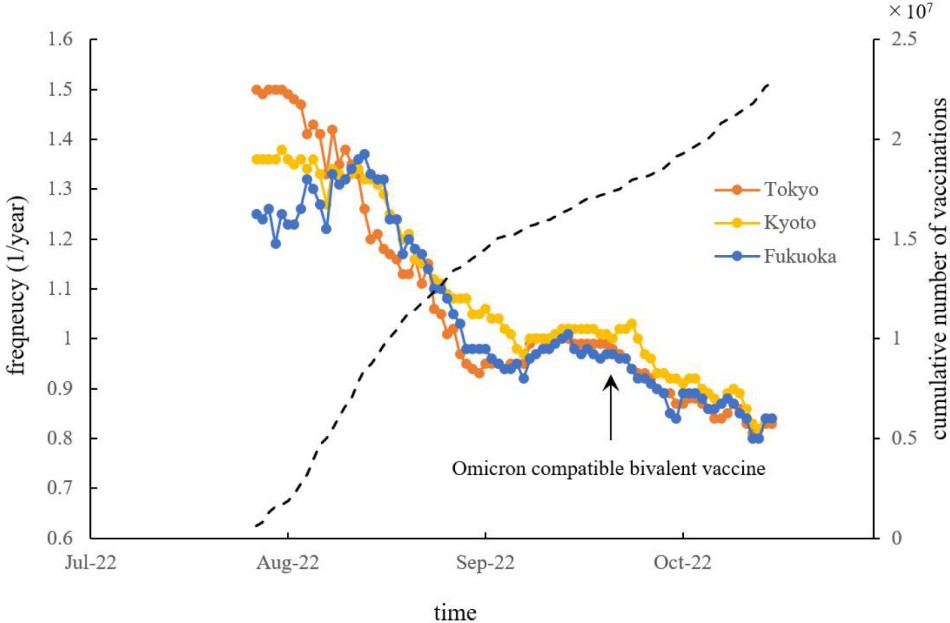

**Fig 11.** Temporal variations of the frequencies of dominant spectral peaks detected in the frequency range of $f \leq 1.6$ for Tokyo (orange), Kyoto (yellow) and Fukuoka (blue) with cumulative numbers of the fourth vaccination, introduced in late-May 2022, and Omicron compatible bivalent vaccine, introduced in mid-September 2022 (dashed line).

data increases from moment to moment, suggests that $f_{1/2}$ in the COVID-19 data for each prefecture corresponds to a subharmonic generated by a period-doubling bifurcation based on a nonlinear dynamical system such as the SEIR model.

The third important result obtained in this study is that the $\lambda$ values of the exponential PSDs increase with population size, as Fig 5 shows, which is comparable to the results obtained for the Danish, USA, and UK communities in our previous study [10]. The magnitude of this $\lambda$-value reflects the magnitude of fluctuations in the data, which have two sources: amplitude fluctuations due to deterministic nonlinearities, and non-deterministic noise, also known as white noise.

### For prefectures with large population sizes (Cluster 1)

All prefectures in Cluster 1 except for Hokkaido possess three characteristics: (i) an exponential PSD, (ii) an identifiable fundamental mode, and (iii) a fundamental mode ($f_1$) and fractional harmonics ($f_{1/2}$) and ($f_2$ and $f_3$) that are observed in multiple segments. These three characteristics (i), (ii) and (iii) were observed in periodic and chaotic time series for the SEIR model, but not in noisy time series in our previous study [9]. Therefore, the large $\lambda$ values for all prefectures except for Hokkaido in Cluster 1 may be due to the occurrence of periodic-doubling bifurcations, resulting in larger power values in the low-frequency region of the PSDs.

### For prefectures with small population sizes (Cluster 2)

In Cluster 2, 76% of the prefectures have spectra with non-subharmonic patterns, in which characteristics (i) and (ii) are observed but characteristic (iii) is not. This result was observed for noisy time series in our previous study [9]. Therefore, the small $\lambda$ values are likely due to the relatively

small power values in the low-frequency region. Regarding the remaining 24% of prefectures whose spectra display subharmonic patterns in Cluster 2 (Fig 7), the λ values of eight of the nine prefectures are above the regression line, i.e., the λ values are relatively large in Cluster 2, and the characteristics (i), (ii) and (iii) are all observed. Thus, in the case of the eight representative prefectures, period-doubling bifurcations may have also occurred, resulting in larger power values in the low-frequency region of the PSD. The reason why period-doubling bifurcations occurred in these eight prefectures, even though they belong to Cluster 2 (small population sizes), as in the case of Cluster 1 (large population sizes), is that, as Fig 8 shows, these eight prefectures except for Miyagi are located in the central part of Japan where the transportation network is well developed and the inflow and outflow of population are large. The situation brought about by the high inflow and outflow of population reflects the characteristics of a nonlinear open system, which guarantees the occurrence of a period-doubling bifurcation.

The following possible factors influencing the prevalence of COVID-19 were considered: 1) immunization [25], 2) population size [26], 3) the emergence of mutant viruses [27], 4) weather [26], 5) societal measures against infectious diseases (e.g., emergency declarations and school closings) [25], 6) infrastructure and cultural background [26] are considered. This study suggests that 1) immunization, 2) population size, and 6) transportation infrastructure may be influencing the prevalence of COVID-19 in Japan since the emergence of the Omicron strain. Continued monitoring is needed to watch for the emergence of new COVID-19 strains and to quantitatively evaluate the impact of factors 1) to 6) on the epidemic variation of COVID-19 by examining the temporal variational structure of epidemic variation, as was done in this study.

The findings of this study highlight the importance of understanding the nonlinear dynamical characteristics of COVID-19's transmission as described by the well-known SEIR mathematical model of infectious diseases, and may have important implications for managing future pandemics as the SEIR model had for measles. Furthermore, this study highlights the need for adaptation strategies that take into account differences in population size (Fig 5) and infrastructure (Fig 8) if Japan is to effectively deal with future pandemics.

To explore the temporal fluctuations of the COVID-19 pandemic, researchers have applied interrupted time series analysis [28, 29] and the autoregressive integrated moving average (ARIMA) model [30]. Another significant technique in time series analysis is the Bayesian spectral estimation method [31]. By contrast, other studies have used the SEIR model [32, 33] to explain the temporal variations of the pandemic. The SEIR model is a well-known nonlinear dynamical system for modeling the spread of infectious diseases [32, 33]. Nevertheless, both interrupted time series analysis and the ARIMA model, which incorporate random noise, as well as the Bayesian spectral estimation method, which assumes approximate linearity in COVID-19 case data, pose challenges in interpreting the multiple periodicities of time series data (Tables 1 and 2) that exhibit unique fluctuations driven by nonlinear dynamics. The 0–1 test employed by Sapkota et al. to analyze the COVID-19 time-series data in Japan [6] is effective for determining whether the data exhibit chaotic behavior. However, this method is not well-suited for detecting multiple periodicities in the time-series data (Tables 1 and 2). In contrast, the current approach, which uses MEM spectral analysis, allows for the identification of periodicities in short time series with high frequency resolution [22]. Notably, segment time-series analysis of short-term data (Figs 10 and 11) revealed that the periodic patterns of the COVID-19 pandemic undergo temporal changes.

In Japan, the number of patients, which was reported on a daily basis up until May 9, 2023, is now reported on a weekly basis. Therefore, an inherent limitation of this study is that the data collected before May 9, which was the subject of the analysis in this study, differs from the data collected after that date in terms of data accuracy, thus limiting the length of data with uniform data accuracy.

## Conclusion

In conclusion, the increase in λ with increasing population size of prefectures in Japan (Fig 5) can be interpreted as follows: as the population size increases, the long-term periodic modes become dominant peaks in the low-frequency ranges of the PSDs due to a period-doubling bifurcation process.

## Supporting information

**S1 Dataset. Time series data of the daily reported number of COVID-19 cases for all 47 prefectures in Japan from 1 February 2022 to 8 May 2023 (714 data points).**
(ZIP)

**S1 Appendix. Assignment of fundamental modes.**
(DOCX)

**S1 Fig. Time series data of the daily reported number of COVID-19 cases for all 47 prefectures in Japan from 1 February 2022 to 8 May 2023.**
(DOCX)

**S2 Fig. MEM-PSDs for the logarithm-transformed data of the daily reported number of COVID-19 cases for all 47 prefectures in Japan from 1 February 2022 to 8 May 2023 ($f <$ 50.0).**
(DOCX)

## Acknowledgments

The author thanks Edanz Group Ltd. (https://jp.edanz.com/ac) for editorial assistance.

## Author Contributions

**Conceptualization:** Ayako Sumi, Masayuki Koyama, Norio Ohtomo.

**Data curation:** Ayako Sumi, Manato Katagiri.

**Formal analysis:** Ayako Sumi, Manato Katagiri.

**Funding acquisition:** Ayako Sumi, Masayuki Koyama.

**Investigation:** Ayako Sumi, Masayuki Koyama, Norio Ohtomo.

**Methodology:** Ayako Sumi, Norio Ohtomo.

**Project administration:** Ayako Sumi, Masayuki Koyama.

**Resources:** Ayako Sumi, Norio Ohtomo.

**Software:** Ayako Sumi, Norio Ohtomo.

**Supervision:** Ayako Sumi.

**Validation:** Ayako Sumi, Masayuki Koyama.

**Visualization:** Ayako Sumi, Manato Katagiri.

**Writing – original draft:** Ayako Sumi.

**Writing – review & editing:** Ayako Sumi, Masayuki Koyama, Norio Ohtomo.

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
