## [Decision Letter · Decision Letter 0]

13 Aug 2024

PONE-D-24-30641Spectral study of COVID-19 pandemic in Japan: the dependence of spectral gradient on the population size of the communityPLOS ONE

Dear Dr. Sumi,

Thank you for submitting your manuscript to PLOS ONE. After careful consideration, we feel that it has merit but does not fully meet PLOS ONE’s publication criteria as it currently stands. Therefore, we invite you to submit a revised version of the manuscript that addresses the points raised during the review process.

We look forward to receiving your revised manuscript.

Kind regards,

Alexander N. Pisarchik, Ph.D.

Academic Editor

PLOS ONE

Journal Requirements:

"Grant Number JP22K10529 JSPS KAKEHI"

3. We note that Figures 1 and 8 in your submission contain map/satellite image which may be copyrighted. All PLOS content is published under the Creative Commons Attribution License (CC BY 4.0), which means that the manuscript, images, and Supporting Information files will be freely available online, and any third party is permitted to access, download, copy, distribute, and use these materials in any way, even commercially, with proper attribution. For these reasons, we cannot publish previously copyrighted maps or satellite images created using proprietary data, such as Google software (Google Maps, Street View, and Earth). For more information, see our copyright guidelines: http://journals.plos.org/plosone/s/licenses-and-copyright.

a. You may seek permission from the original copyright holder of Figures 1 and 8 to publish the content specifically under the CC BY 4.0 license.  

4. We are unable to open your Supporting Information files [from Short_1_1_COVID19_Hokkaido_230518.dn to Short_1_48_COVID19_ALL_230518.dn]. Please kindly revise as necessary and re-upload.

Additional Editor Comments:

Please address the reviewers' comments and revise the paper accordingly. A final decision on the publication will be made following the second round of revisions.

Reviewers' comments:

Reviewer's Responses to Questions

**Comments to the Author**

1. Is the manuscript technically sound, and do the data support the conclusions?

Reviewer #1: Yes

Reviewer #2: Partly

2. Has the statistical analysis been performed appropriately and rigorously? 

Reviewer #1: Yes

Reviewer #2: No

3. Have the authors made all data underlying the findings in their manuscript fully available?

Reviewer #1: Yes

Reviewer #2: Yes

4. Is the manuscript presented in an intelligible fashion and written in standard English?

Reviewer #1: Yes

Reviewer #2: Yes

5. Review Comments to the Author

Reviewer #1: The paper examines the statistical data of COVID-19 across 47 prefectures in Japan, aiming to explain the epidemic's progression through the lens of previously considered dynamical models known for their complex attractors and non-trivial effects. The topic is undeniably intriguing and warrants thorough exploration. The research is well-motivated, its relevance and significance driven by the pressing need for highly predictive epidemic models, as evidenced by the three key findings outlined in the Discussion section.

However, several questions and comments arise:

1. What are the limitations inherent to this study?

2. Could the methodologies employed be extrapolated to sub-regions within other countries, encompassing diverse geographical and societal contexts like India, China, Turkey, or less densely populated regions such as Argentina, Canada, and Kazakhstan? Is population density and transport infrastructure the primary driver, or do "cultural features" play a significant role? What do you think about it?

3. Are there any factors, beyond those explicitly considered, that could potentially influence the study's results or interpretations?

4. What implications do these findings hold for comprehending and effectively combating future pandemics?

5. Would logarithmic scales be more effectively represented using scientific notation (e.g., 10^0, 10^1, ..., 10^n, 10^{n+1}, ...) for the major ticks with the values condensed towards the right for minor ticks, rather than simply log_10 exponents? I think, this would enhance visual clarity.

6. The figures, while informative, appeared over-compressed and of low quality in my version of the document. I trust the final publication will employ a resolution of at least 300 dpi to ensure optimal visual clarity.

Reviewer #2: Review report

Draft title: Spectral study of COVID-19 pandemic in Japan: the dependence of spectral gradient on the population size of the community

The authors of the reviewed draft show a dynamic analysis of COVID 19 in 47 prefectures in Japan.

1. The authors don’t mention if the studied series considers people who had received a previous vaccine for Omricon previous strain.

2. All the information was analyzed for the unique data series, and that information was studied as an absolute truth, without comparing with other results to mention the validity of the analysis

3. I’m unsure what the “lambda” parameter means; I looked for the definition but didn’t find it.

4. In general, all the figures need to be presented in a higher quality.

5. This type of work shows analysis of temporal series by using different mathematical tools, but it isn’t accomplished with the scientific method characteristics. Of course the results couldn’t be reproduced again but authors could propose a more serious analysis and scientific content.

I don’t consider that the presented draft could be published in PLUS ONE Journal.

6. PLOS authors have the option to publish the peer review history of their article (what does this mean?). If published, this will include your full peer review and any attached files.

Reviewer #1: No

Reviewer #2: No

---

## [Author Response · Author response to Decision Letter 0]

1 Nov 2024

Responses to comments 

In response to the referees' comments, the following points have been changed in the revised manuscript. These changes are highlighted in bold type. The new version of the manuscript has been edited by professional editors at Edanz Group Ltd. 

Response to Reviewer 1 

Comment 1: What are the limitations inherent to this study?

Response: In Japan, the number of patients, which was reported on a daily basis up until May 9, 2023, is now reported on a weekly basis. Therefore, an inherent limitation of this study is that the data collected before May 9, which was the subject of the analysis in this study, differs from the data collected after that date in terms of data accuracy, thus limiting the length of data with uniform data accuracy. To present this limitation, the following sentences have been added to the new manuscript (page 34, lines 461–465). 

“In Japan, the number of patients, which was reported on a daily basis up until May 9, 2023, is now reported on a weekly basis. Therefore, an inherent limitation of this study is that the data collected before May 9, which was the subject of the analysis in this study, differs from the data collected after that date in terms of data accuracy, thus limiting the length of data with uniform data accuracy.”

Comment 2: Could the methodologies employed be extrapolated to sub-regions within other countries, encompassing diverse geographical and societal contexts like India, China, Turkey, or less densely populated regions such as Argentina, Canada, and Kazakhstan? Is population density and transport infrastructure the primary driver, or do "cultural features" play a significant role? What do you think about it?

Response: We appreciate your valuable comments. We believe that your main comments can be summarized as the following two points.

1) Could the methodologies employed be extrapolated to sub-regions within other countries, encompassing diverse geographical and societal contexts like India, China, Turkey, or less densely populated regions such as Argentina, Canada, and Kazakhstan?

2) Is population density and transport infrastructure the primary driver, or do "cultural features" play a significant role? What do you think about it?

To answer the first point, the method used for analysis in this study is capable of examining the structure of temporal variation even for short data lengths and can be applied to any time series with at least seven data points, e.g., surveillance data from countries with geographical and social backgrounds different to those of Japan. To demonstrate the applicability of this method to short data lengths, the following new sentences and three references have been added to the revised manuscript (page 9, lines 135–138, reference numbers 21, 22 and 22).

“The method used for analysis in this study can be applied to any time series with at least seven data points [20, 21]. For example, it has been applied to time series data of sunspot numbers [22], which were used as a test case for time series analysis, and its usefulness has been confirmed.”

[20] Ohtomo N, Tokiwano K, Tanaka Y, Sumi A, Terachi S, Konno H. Exponential Characteristics of Power Spectral Densities Caused by Chaotic Phenomena. J Phys Soc Jpn 1995; 64(4): 1104-1113.

[21] A Recent Advance of Biological Time Series Analysis–Maximum Entropy Method and its Applications to Medical and Biological Science. Sapporo, Hokkaido University Press. 1994.

[22] Ohtomo N, Terachi S, Tanaka Y, Tokiwano K, Kaneko N. New method of time series analysis and its application to Wolf’s sunspot number data. Jpn J Appl Phys. 1994; 33: 2821–2831.

 As to the second point, to address the possible factors influencing the prevalence of COVID-19, the following sentences were added to the revised manuscript (page 32, lines 424–434) with new three references (reference numbers 25, 26, and 27).

“The following possible factors influencing the prevalence of COVID-19 were considered: 1) immunization [25], 2) population size [26], 3) the emergence of mutant viruses [27], 4) weather [26], 5) societal measures against infectious diseases (e.g., emergency declarations and school closings) [25], 6) infrastructure and cultural background [26] are considered. This study suggests that 1) immunization, 2) population size, and 6) transportation infrastructure may be influencing the prevalence of COVID-19 in Japan since the emergence of the Omicron strain. Continued monitoring is needed to watch for the emergence of new COVID-19 strains and to quantitatively evaluate the impact of factors 1) to 6) on the epidemic variation of COVID-19 by examining the temporal variational structure of epidemic variation, as was done in this study.”

[25] Du H, Saiyed S, Gardner ML. Association between vaccination rates and COVID-19 health outcomes in the United States: a population-level statistical analysis. BMC Public Health. 2024; 24: 220–233. 

[26] Kong JD, Tekwa EW, Gignoux-Wolfsohn SA. (2021). Social, economic, and environmental factors influencing the basic reproduction number of COVID-19 across countries. PLoS One. 2021; 16(6): e0252373. 

[27] World Health Organization. Q&A: COVID-19 variants and what they mean for countries and individuals. [accessed 2024 September 15]. Available from: https://www.who.int/europe/news/item/20-05-2021-q-a-covid-19-variants-and-what-they-mean-for-countries-and-individuals.

Comment 3: Are there any factors, beyond those explicitly considered, that could potentially influence the study's results or interpretations?

Response: Answer to this question overlaps with the answer to the second point of Question 2 described above, but factors potentially influencing the spread of COVID-19 include 1) vaccination efforts [25], 2) the size of the population [26], 3) the emergence of viral variants [27], 4) climatic conditions [26], 5) public health measures (such as lockdowns and school closures) [25], and 6) infrastructure and cultural factors [26]. This study indicates that vaccination, population size, and transportation infrastructure might have played a role in shaping the prevalence of COVID-19 in Japan, particularly since the Omicron variant emerged. 

Comment 4: What implications do these findings hold for comprehending and effectively combating future pandemics?

Response: The findings of this study highlight the importance of understanding the nonlinear dynamical characteristics of COVID-19’s transmission, as described by the well-known SEIR mathematical model of infectious diseases, and may have important implications for managing future pandemics, as the SEIR model had for measles. Furthermore, this study highlights the need for adaptation strategies that take into account differences in population size (Fig 5) and infrastructure (Fig 8) if Japan is to effectively deal with future pandemics. To emphasize this point, the following sentences have been added to the revised manuscript (page 32, lines 435–441).

“The findings of this study highlight the importance of understanding the nonlinear dynamical characteristics of COVID-19’s transmission as described by the well-known SEIR mathematical model of infectious diseases, and may have important implications for managing future pandemics as the SEIR model had for measles. Furthermore, this study highlights the need for adaptation strategies that take into account differences in population size (Fig 5) and infrastructure (Fig 8) if Japan is to effectively deal with future pandemics.”

Comment 5: Would logarithmic scales be more effectively represented using scientific notation (e.g., 10^0, 10^1, ..., 10^n, 10^{n+1}, ...) for the major ticks with the values condensed towards the right for minor ticks, rather than simply log_10 exponents? I think, this would enhance visual clarity.

Response: According to this suggestion, we have revised Figs 5 and 7. 

Comment 6: The figures, while informative, appeared over-compressed and of low quality in my version of the document. I trust the final publication will employ a resolution of at least 300 dpi to ensure optimal visual clarity.

Response: We apologize for the poor resolution of the previously submitted figures. In the revised version, we have increased the resolution of the figures to at least 300 dpi.

Response to Reviewer 2 

Comment 1: The authors don’t mention if the studied series considers people who had received a previous vaccine for Omricon previous strain.

Response: To address this comment, we have added the following sentences regarding people who had received the four dose of the COVID-19 vaccine and a vaccine for the Omicron strain in the Materials and Methods section (page 7, lines 93–105) and have added new references (reference numbers 13 and 14).

“Regarding vaccines administered during the period of data analyzed in this study, the initial vaccination efforts in Japan began in February 2021, focusing first on healthcare workers and the elderly aged 65 and above. By 2024, the first-dose vaccination rate had reached 80.4% [13], while the second and third doses were administered to bolster immunity against the spread of the Alpha and Delta variants. These booster doses were crucial in maintaining protection as new variants emerged [13].

The fourth dose of the COVID-19 vaccine, initiated on May 25, 2022, primarily targeted individuals aged 60 and older along with those with underlying health conditions, achieving a vaccination rate of approximately 54% by early 2023 [13]. Meanwhile, the bivalent Omicron-adapted vaccine, introduced in the autumn of 2022, was rapidly expanded to broader age groups, resulting in a total of around 28.4 million doses that had been administered by 2023 [14].” 

[13] Ministry of Health, Labour and Welfare. Information on COVID-19 Vaccines. [accessed on 2024 September 30] Available from: https://www.mhlw.go.jp/stf/seisakunitsuite/bunya/vaccine_shingata.html.

[14] National Institute of Infectious Diseases (NIID). COVID-19 Related Information Page. [accessed on 2024 September 30] Available from: https://www.niid.go.jp/niid/ja/diseases/ka/covid19.html.

Comment 2: All the information was analyzed for the unique data series, and that information was studied as an absolute truth, without comparing with other results to mention the validity of the analysis

Response: We believe that your main comments can be summarized by the following two points.

1) The validity of obtained results should be addressed.

2) The validity of the present method for time series analysis should be addressed.

To address the first point, data validity can be examined by checking the data against the results of the mathematical model. In this study, as described in the Introduction (lines 75–76) and Discussion (lines 382–391 and 401–403), the analysis results of this study were checked against the well-known SEIR mathematical model.　To make it clearer that we perform this comparison, we have added the following text to the Abstract (page 3, lines 28–29 and 34–35).

Line 28–29: “…, such as the susceptible/exposed/infectious/recovered (SEIR) epidemic model.”

Line 34–35: “…, as predicted by theoretical studies based on the SEIR model.”

As to the second point, the method has been applied to time series data of sunspot numbers, which were used as a test case for time series analysis, and its usefulness has been confirmed in a study reported by one of the authors of this study (Ohtomo N). To make it clear that we are doing this comparison, we have added the following sentences to the Materials and Methods (page 9, lines 135–138) and new references (reference numbers 20, 21 and 22).

“The method used for analysis in this study can be applied to any time series with at least seven data points [20, 21]. For example, it has been applied to time series data of sunspot numbers [22], which were used as a test case for time series analysis, and its usefulness has been confirmed.”

[20] Ohtomo N, Tokiwano K, Tanaka Y, Sumi A, Terachi S, Konno H. Exponential Characteristics of Power Spectral Densities Caused by Chaotic Phenomena. J Phys Soc Jpn 1995; 64(4): 1104-1113.

[21] A Recent Advance of Biological Time Series Analysis –Maximum Entropy Method and its Applications to Medical and Biological Science. Sapporo,

[22] Ohtomo N, Terachi S, Tanaka Y, Tokiwano K, Kaneko N. New method of time series analysis and its application to Wolf’s sunspot number data. Jpn J Appl Phys. 1994; 33: 2821–2831.

Comment 3: I’m unsure what the “lambda” parameter means; I looked for the definition but didn’t find it.

Response: To address this comment, the following sentences have been added to the revised manuscript (pages 14–15, lines 219–224).

“To obtain the magnitude of λ, the mean power of the PSD was calculated by integrating the PSD over a small frequency interval Δf, that is, the mean power of the PSD is the power in the interval of frequencies [f, f +Δf]. The line of the PSD gradient is calculated as a regression line against the mean powers in the low frequency range (f < 12.0), as illustrated in Fig 4. The precise value of λ was determined using this procedure.” 

Comment 4: In general, all the figures need to be presented in a higher quality.

Response: We apologize for the poor resolution of the previously submitted figures. In the revised version, we have increased the resolution of the figures to at least 300 dpi.

Comment 5: This type of work shows analysis of temporal series by using different mathematical tools, but it isn’t accomplished with the scientific method characteristics. Of course the results couldn’t be reproduced again but authors could propose a more serious analysis and scientific content.

Response: To compare the methodology and result in the current study with other approaches in this topic, we have added a new sentence in the Discussion section (pages 33–34, lines 442–453) and eight new references (reference number 28–35), as follows:

“To explore the temporal fluctuations of the COVID-19 pandemic, researchers have applied interrupted time series analysis [28, 29] and the autoregressive integrated moving average (ARIMA) model [30]. Another significant technique in time series analysis is the Bayesian spectral estimation method [31]. By contrast, other studies have used the SEIR model [32, 33] to explain the temporal variations of the pandemic. The SEIR model is a well-known nonlinear dynamical system for modeling the spread of infectious diseases [34, 35]. Nevertheless, both interrupted time series analysis and the ARIMA model, which incorporate random noise, as well as the Bayesian spectral estimation method, which assumes approximate linearity in COVID-19 case data, pose challenges in interpreting the multiple periodicities of time series data (Tables 1 and 2) that exhibit unique fluctuations driven by nonlinear dynamics.”

New references:

[28] Vokó Z, Pitter JG. The effect of social distance measures on COVID-19 epidemics in Europe: an interrupted time series analysis. GeroScience 2020; 42: 1075-7082.

[29] Islam N, Sharp SJ, Chowell G, Shabnam S, Kawachi I, Lacey B, et al. 26. Physical distancing interventions and incidence of coronavirus disease 2019: natural experiment in 149 countries. BMJ 2020; 370: m27431.

[30] Chyon FA, Suman NH, Fahim RI, Ahmmed S. Time series analysis and predicting COVID-19 affected patients by ARIMA model using machine learning. J Virol Methods 2022; 301: 114433.

[31] Nason G. COVID-19 cycles and rapidly evaluating lockdown strategies using spectral analysis. Sci Rep 2020; 10: 22134. 

[32] He S, Peng Y, Sun K. SEIR modeling of the COVID-19 and its dynamics. Nonlinear Dyn. 2020; 101: 1667-1680.

[33] Wang A, Jamal SS, Yang B, Pham VT. Complex behavior of COVID-19’s mathematical model. Eur Phys J Spec Top 2022; 231: 885-891.

[34] He S, Peng Y, Sun K. SEIR modeling of the COVID-19 and its dynamics. Nonlinear Dyn. 2020; 101: 1667-1680.

[35] Wang A, Jamal SS, Yang B, Pham VT. Complex behavior of COVID-19’s mathematical model. Eur Phys J Spec Top 2022; 231: 885-891.

With respect to other approaches of investigating Japanese pandemic pattern, we have added a new sentence in the Discussion section (page 34, lines 453–460) as follows:

“The 0-1 test employed by Sapkota et al. to analyze the COVID

---

## [Decision Letter · Decision Letter 1]

7 Nov 2024

Spectral study of COVID-19 pandemic in Japan: the dependence of spectral gradient on the population size of the community

PONE-D-24-30641R1

Dear Dr. Sumi,

We’re pleased to inform you that your manuscript has been judged scientifically suitable for publication and will be formally accepted for publication once it meets all outstanding technical requirements.

Kind regards,

Alexander N. Pisarchik, Ph.D.

Academic Editor

PLOS ONE

Additional Editor Comments (optional):

Reviewers' comments:

Reviewer's Responses to Questions

**Comments to the Author**

1. If the authors have adequately addressed your comments raised in a previous round of review and you feel that this manuscript is now acceptable for publication, you may indicate that here to bypass the “Comments to the Author” section, enter your conflict of interest statement in the “Confidential to Editor” section, and submit your "Accept" recommendation.

Reviewer #1: All comments have been addressed

Reviewer #2: All comments have been addressed

2. Is the manuscript technically sound, and do the data support the conclusions?

Reviewer #1: Yes

Reviewer #2: Yes

3. Has the statistical analysis been performed appropriately and rigorously? 

Reviewer #1: Yes

Reviewer #2: Yes

4. Have the authors made all data underlying the findings in their manuscript fully available?

Reviewer #1: Yes

Reviewer #2: Yes

5. Is the manuscript presented in an intelligible fashion and written in standard English?

Reviewer #1: Yes

Reviewer #2: Yes

6. Review Comments to the Author

Reviewer #1: The Authors made great work to upgrade the manuscript.

All my comments have been substantially addressed.

I recommend to accept the paper.

Reviewer #2: The authors have answered the questions and have improved the writing of the document, in that sense the work could be considered for publication if the editor so considers it.

7. PLOS authors have the option to publish the peer review history of their article (what does this mean?). If published, this will include your full peer review and any attached files.

Reviewer #1: No

Reviewer #2: No

---

## [Editor Report · Acceptance letter]

12 Nov 2024

PONE-D-24-30641R1 

PLOS ONE

Dear Dr. Sumi, 

I'm pleased to inform you that your manuscript has been deemed suitable for publication in PLOS ONE. Congratulations! Your manuscript is now being handed over to our production team.

Kind regards, 

on behalf of

Prof. Alexander N. Pisarchik 

Academic Editor

PLOS ONE